# The Use of the DRASTIC-LU/LC Model for Assessing Groundwater Vulnerability to Nitrate Contamination in Morogoro Municipality, Tanzania

Neema J. Mkumbo [1,2,*], Kassim R. Mussa [2], Eliapenda E. Mariki [3] and Ibrahimu C. Mjemah [2]

1  Department of Geology, University of Dodoma, Dodoma P.O. Box 11090, Tanzania
2  Department of Geography and Environmental Studies, Sokoine University of Agriculture, Morogoro P.O. Box 3038, Tanzania
3  Department of Chemistry and Physics, Sokoine University of Agriculture, Morogoro P.O. Box 3038, Tanzania
*  Correspondence: neema.mkumbo@udom.ac.tz; Tel.: +255-629100751

**Abstract:** Groundwater is a useful source of water for various uses in different places. The major challenge in the use of this resource is how to manage and protect it from contamination. The current study was conducted in Morogoro Municipality to identify vulnerable groundwater areas by using DRASTIC-LU/LC model. The study applied eight input parameters, i.e., depth to water table, net recharge, aquifer media, soil media, topography, impact of vadose zone, hydraulic conductivity and land use/land cover patterns, which were overlaid in GIS to generate groundwater vulnerable map. The model used rating (R = 1–10) and weighting (W = 1–5) techniques to assess the effect of each parameter on groundwater contamination. The DRASTIC-LU/LC Vulnerability Index map was classified into low- (area = 29.2 km$^2$), moderate- (area = 120.4 km$^2$) and high-vulnerability zones (area = 124.4 km$^2$). Nitrate analysis was conducted using the cadmium reduction method (DR 890) to assess the validity of the model and it was observed that 55%, 15% and 50% of the samples with unacceptable (>50 mg/L), high (29–50 mg/L) and moderate (14–28 mg/L) nitrate concentrations, respectively, fall into the high-vulnerability zone. Furthermore, 45%, 70% and 50% of the samples with unacceptable, high and moderate nitrate concentrations, respectively, fall into the moderate-vulnerability zone. In the low-vulnerability zone, only 15% of samples were found with a high nitrate concentration.

**Keywords:** groundwater; DRASTIC-LU/LC model; nitrate contamination; vulnerability map





## 1. Introduction

Groundwater is a useful source of water for different socio-economic activities, and it contributes to about 99% of freshwater reserves worldwide [1], which can be easily detected by the presence of rivers, lakes, springs and wetlands. Groundwater is the most trusted water source in different countries of the world. In India, for example, groundwater contributes to about 80% and 50% of domestic water in rural and urban areas, respectively [2]. In Tanzania, groundwater provides a potential source of fresh water, and it has been utilized for different purposes, such as domestic uses (60%) in both urban and rural areas, agriculture (10%), industrial and mining (2%) and other uses, including livestock and dry land fishing (28%) [3].

The use of groundwater faces various challenges, one of them being contamination due to natural and human activities. In some areas, human activities, such as the dumping of waste materials and sewage, agriculture and industrial activities, have been associated with groundwater contamination [4–7]. In addition, geological and hydrogeological factors may accelerate the magnitude of groundwater contamination within a particular area [7].

The concept of groundwater contamination vulnerability is based on the theory of diffusion, infiltration and percolation of the contaminant species from the Earth's surface

to the aquifer system [8]. The diffusion and infiltration processes are highly influenced by hydrogeological factors and human impact during land uses. The attenuation of contaminants into the ground involves physical, chemical and biological processes between the contaminant species and the media through which they pass [9]. Due to the heterogeneity of hydrogeological parameters such as soil and aquifer materials, which result in differences in conductivity and transmissivity, the magnitude and levels of contamination tend to vary from one place to another.

Nitrate has been documented as the major contaminant in groundwater since it is highly soluble in water, and it has been a growing environmental problem worldwide [2,10,11]. However, the vulnerability of aquifers to such pollutant has not received considerable scientific attention. According to previous studies [12], nitrate contamination in groundwater in Tanzania has been a challenge for years. Accordingly, nitrate concentrations were recorded in different cities in Tanzania and were found to be above the permissible limit (50 mg/L). These cities are Dar es Salaam (477.6 mg/L), Dodoma (441.1 mg/L), Tanga (100 mg/L), Manyara (180 mg/L) and Arusha (>50 mg/L). The concentration of nitrate in groundwater in Morogoro municipality has been recorded to be 32 mg/L [13], while in the present study, the concentration was recorded at 284.1 mg/L. Additionally, the factors contributing to nitrate in groundwater in most places in the country remain poorly documented. A high concentration of nitrate (>50 mg/L) in groundwater may lead to negative impacts on human health, such as the "blue baby" syndrome in babies aged less than one year, spontaneous abortion, thyroid disorder and stomach cancer [14]. The control and prevention of nitrate contamination require knowledge of the level and sources. The factors influencing groundwater contamination and nitrate concentrations in groundwater in Tanzania, particularly in the Morogoro municipality, remain poorly understood. Furthermore, in most regions within the country, including the study area, there is missing information on which areas are more vulnerable to groundwater contamination.

Many researchers elsewhere have applied different models to establish vulnerable groundwater zones. These include overlay and index models [4,5,10,15–17], process-based models [9,18] and statistical models [19–21]. The common working principle of these models is taking into account the chemical, physical and biological processes that take place between the aquifer media and the contaminants [16]. Thus, this study uses the DRASTIC-LU/LC model, a modified DRASTIC model with overlay and index models to delineate groundwater areas vulnerable to nitrate contamination. This model has an advantage over the generic DRASTIC model since it considers the contribution of land use/land cover (LU/LC) patterns towards groundwater contamination, unlike the conventional DRASTIC model, which considers only hydrogeological factors [22]. Generally, the DRASTIC-LU/LC model applies the additive mathematical formulation of eight parameters that fall into five categories, such as geological (aquifer media, soil media and impact of vadose zone), hydrogeological (depth to groundwater level and hydraulic conductivity), geomorphological (topography), meteorological (net recharge) and anthropogenic (land use). Each parameter has a weight and rating assigned according to its susceptibility to groundwater contamination; then, the summation of all factors is considered to create a groundwater vulnerability index map. The main objective of this study is to delineate vulnerable groundwater zones to nitrate contamination in the Morogoro municipality through the DRASTIC-LU/LC model.

## 2. Materials and Methods

### 2.1. Description of the Study Area

2.1.1. Location

Morogoro municipality (Figure 1) is one of the districts in the Morogoro region found on the west of the coast of Tanzania, about 196 km west of Dar es Salaam city and 260 km east of Dodoma, the capital city of Tanzania. It covers an area of 274 km$^2$ and it is bounded by the Mvomero and Morogoro districts. The land use type in the study area includes residential use, both planned and unplanned, in some places in hand with gardening,

institutional use, agricultural use, forestry, commercial use, water bodies and open spaces, such as playgrounds, golf courses and others. Residential and agricultural land use patterns are considered to be associate with groundwater contamination since they facilitate the production of contaminants, for example, the release of nitrogen compounds from manure and fertilizers as well as from sewage.

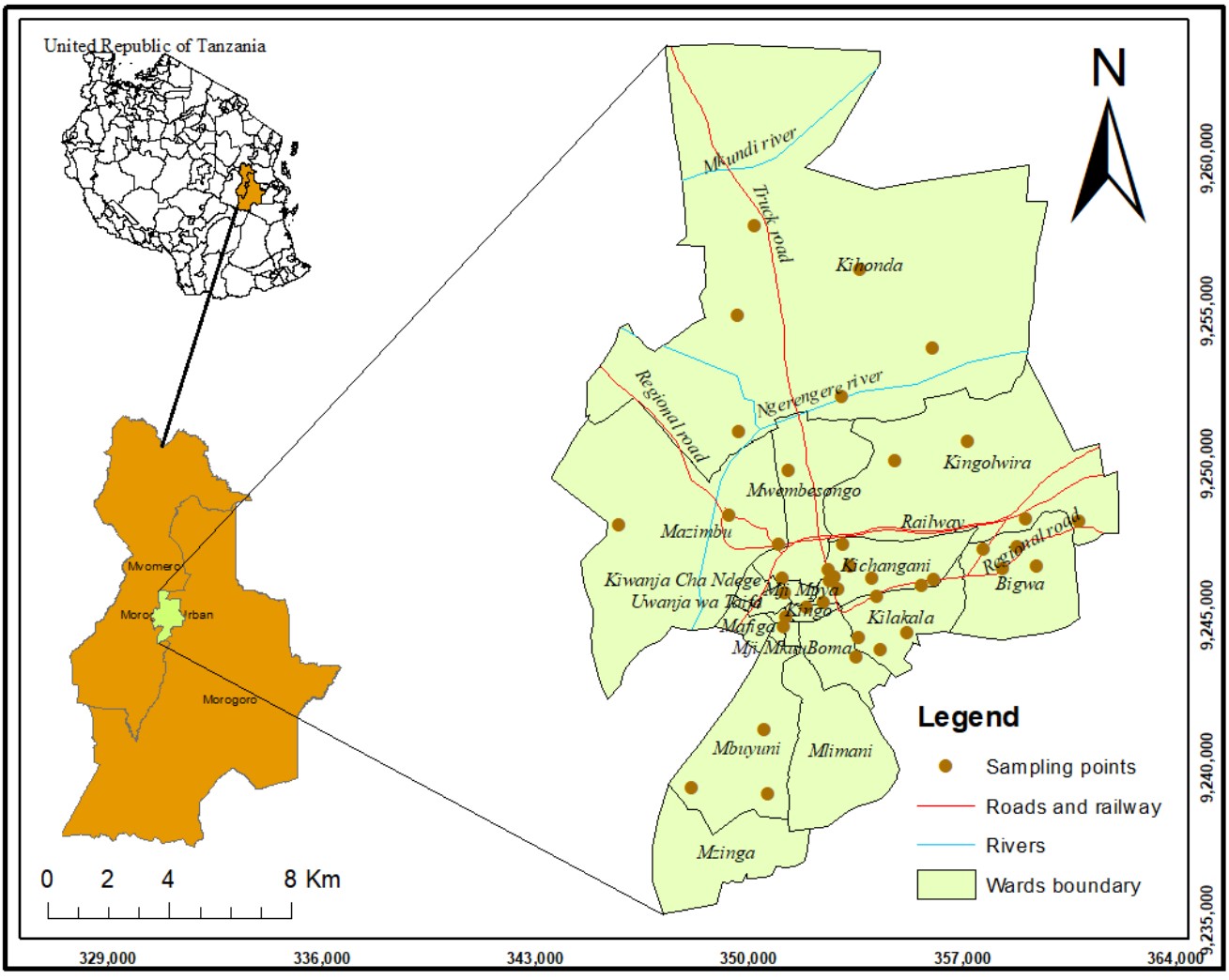

**Figure 1.** Location of the study area.

### 2.1.2. Climate

The study area receives a bimodal rainfall, starting from October to January with a short rain period, followed by a relatively short dry season in February. From March to May, a long rain period begins, with a total rainfall of 821 mm to 1050 mm, followed by a long dry season from June to September [23]. Generally, the area has an average annual rainfall that varies between 600 mm and 1800 mm and an average annual minimum and maximum temperature that varies between 18 °C and 30 °C [23,24]. In addition, this is substantiated by climatology data recorded from 2007 to 2021 (Figure 2).

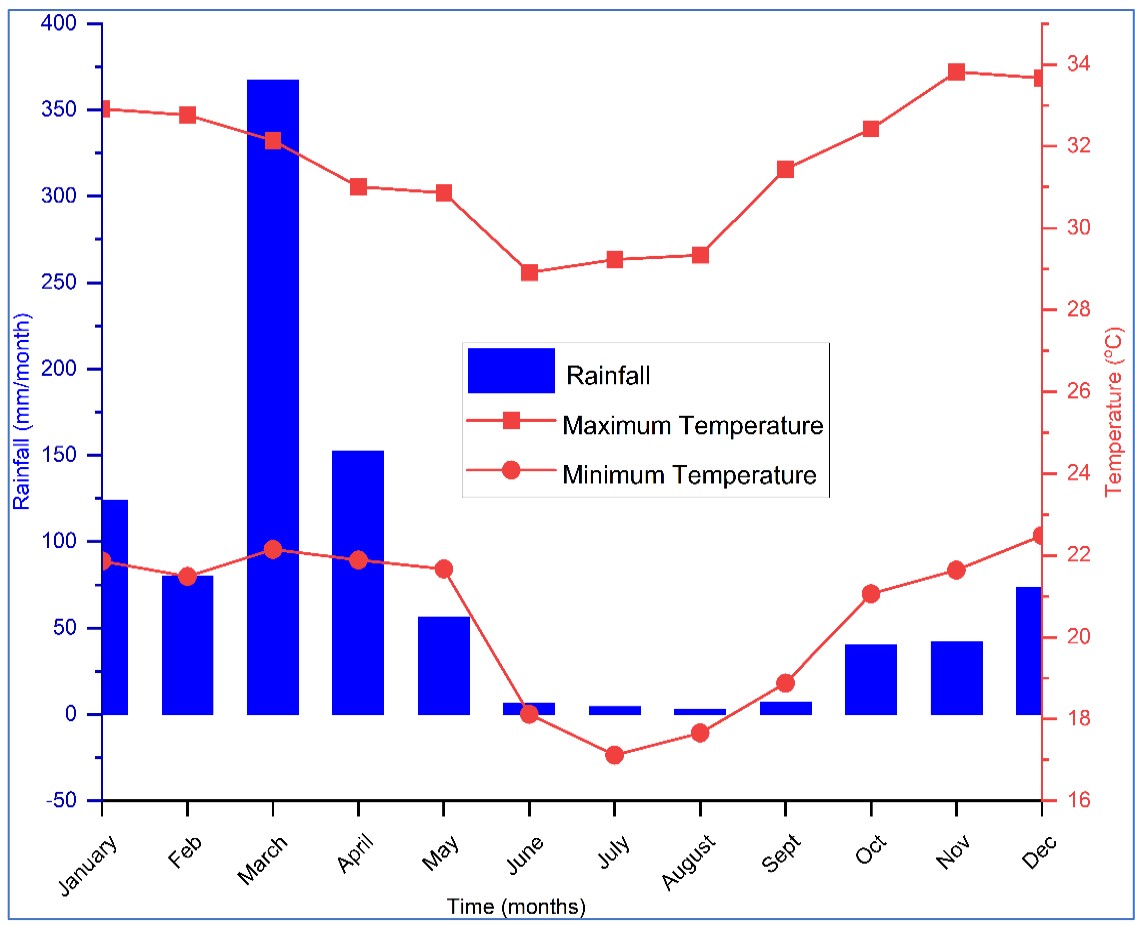

**Figure 2.** Monthly average rainfall and maximum and minimum temperature from 2007 to 2022 (Source: Tanzania Meteorological Agency-Morogoro).

### 2.1.3. Hydrology

Morogoro municipality is a part of the Ngerengere catchment, which originates from the Uluguru Mountains and is located in the middle of the Wami–Ruvu basin. The Ngerengere river is fed by four tributaries, namely Mgeta river, Mlali river, Mzinga river and Lukumeni river, which also originate from the Uluguru Mountains [25]. The major surface water source in the municipality is the Mindu dam, which receives water from three major river tributaries originating from Kasanga hills [24,25].

### 2.1.4. Geology and Hydrogeology

Regionally, the Morogoro municipality is made up of a Precambrian basement complex known as the Usagaran unit, characterized by high-grade metamorphic rocks such as amphibolite, gneiss and granulites, and a Neogene formation characterized by a thick deposit of red soil, "mbuga" soil and alluvium, as shown in Figure 3. The alluvial formation is the result of river deposition process as it is found around the river system. The porosity and permeability are relatively high in alluvial and sand aquifers compared to the metamorphic formation, where secondary porosity is common. Generally, the Neogene formation consists of the major aquifer of the Morogoro Municipality as it was found that many wells are located in such formation, while few samples were found in the fractured aquifer (metamorphic rocks). The aquifer of the study area varies in thickness from 5 m to 50 m and transmissivity and conductivity ranging from 2.3 to 9.7 $m^2$/day and 0.14 to 4.3 m/day. This is according to the Drilling and Dam Construction Agency data of 2017–2020.

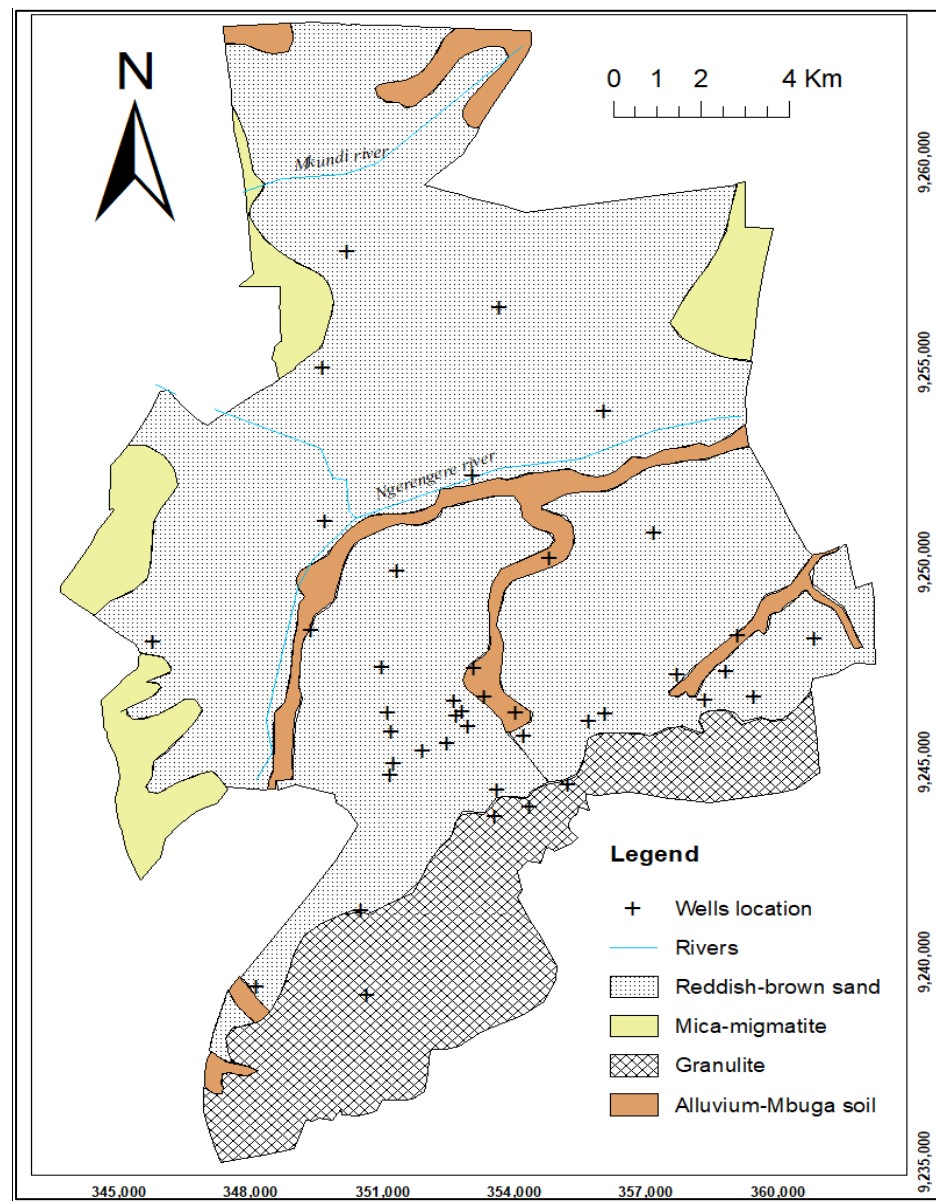

**Figure 3.** Hydrogeological map of the study area.

*2.2. Mapping of DRASTIC-LU/LC Parameters*

2.2.1. Depth to the Water Table

This is the distance from the Earth's surface to the groundwater level. It has an impact on groundwater contamination since it determines the time of travel of contaminants from the surface to groundwater. This means that the lower the depth, the shorter the travel time and, hence, the higher the possibility of groundwater contamination and vice versa. In the present study, the depth to the water table of wells (about 62% of all wells) was measured using a dipper, and the coordinates were taken by Global Positioning System (GPS) device. However, due to the sealing of some wells (38% of all wells) that made it difficult to take measurement directly, the recorded static water level data for 2020–2021 were acquired from the Drilling and Dam Construction Agency (DDCA). The water level data and their locations were overlaid in Geographical Information System (GIS) software and then interpolated by using Inverse Distance Weighting (IDW) function under the spatial analyst tool. The weight of 5 was assigned to this parameter, and the rating was provided with modification from [6,22], as shown in Table 1.

**Table 1.** Weight and rating of the DRASTIC-LU/LC parameters.

| Parameter | Parameter Ranges | Weight | Rating |
|---|---|---|---|
| DTWT (m) | 0.6–4.9 | 5 | 10 |
| | 4.9–7.5 | | 9 |
| | 7.5–9.5 | | 7 |
| | 9.5–11.9 | | 5 |
| | 11.9–15.2 | | 3 |
| Net recharge (mm/y) | 157–198 | 4 | 5 |
| | 198–228 | | 6 |
| | 228–270 | | 7 |
| | 270–317 | | 8 |
| | 317–369 | | 9 |
| | Sand | | 8 |
| Aquifer media | Alluvium | 3 | 6 |
| | Granulite | | 3 |
| Soil media | Loamy sand | 2 | 8 |
| | Silty clay | | 3 |
| Topography (%) | 0–2 | 1 | 10 |
| | 2–6 | | 9 |
| | 6–12 | | 5 |
| | 12–18 | | 3 |
| | >18 | | 1 |
| I. of vadose zone | Sand with clay | 5 | 4 |
| | Granitic gneiss | | 6 |
| | Sand and gravel | | 8 |
| H. conductivity (m/day) | 0.1–0.8 | 3 | 1 |
| | 0.8–1.4 | | 4 |
| | 1.4–2.1 | | 6 |
| | 2.1–3.0 | | 8 |
| | 3.0–4.3 | | 10 |
| LU/LC | Settlement | 5 | 10 |
| | Agriculture | | 8 |
| | Water bodies | | 5 |
| | Vegetation cover | | 3 |

### 2.2.2. Net Recharge

This is the volume of water that infiltrates into the ground to create groundwater, which may carry contaminants to the aquifer. The larger the volume of water into the aquifer, the larger the capacity to carry a large amount of contaminants. The amount of recharge is directly proportional to the risk of contamination, until the saturation point is reached, where the addition of water results in dilution and hence decreases the quantity of contaminants.

The net recharge of the study area was estimated using two methods, namely the Curve Number method, a recharge estimation method using Thornthwaite water balance software [26] and Rao relationship equation [27,28]. With the first method, daily maximum and minimum temperature from Tanzania Meteorological Agency (TMA) were utilized to calculate Potential Evapotranspiration (PET) by using the Hargreaves–Samani method. Curve number was deduced from LULC map. Thus PET, CN, precipitation and other parameters such as soil moisture and plant available water (PAW) obtained from www.northeastcropadvisers.org (accessed on 15 June 2022) were used as inputs in the Thornthwaite software. In the second method (Rao relationship equation), only the annual precipitation data are required, as shown in Equation (1).

$$R = 0.35 \, (P - 600) \text{ for areas with precipitation above 1000 mm/year} \tag{1}$$

where R and P are recharge and precipitation, respectively, expressed in millimeters (mm). According to [28], the Rao relationship equation is suitable in areas where the soil type is not studied at a small scale. This condition is similar to the case in this study area.

The calculated recharge by the Thornthwaite recharge estimation method and Rao relationship equation was in the range of 160.4–378 mm/year and 174.91–361.16 mm/year, respectively. Since there is a little deviation in the recharge estimated by the two methods, the average recharge was considered. The recharge data overlaid in ArcGIS and interpolated using IDW, and the weight of 4 was assigned, with the rating modified from [6,22] as shown in Table 1.

### 2.2.3. Aquifer Media

These are the aquifer materials that are either consolidated or unconsolidated. The grain size on aquifer materials defines the aquifer porosity and impact on the groundwater quality. Coarse-grained aquifer materials result in high porosity and permeability, which also eases the infiltration and percolation of contaminants into the aquifer. On the other hand, fine-grained aquifer materials impose infiltration and percolation resistance to the aquifer. The aquifer media of the study area is made up of Neogene formation, characterized by a thick deposit of red soil, "mbuga" soil and alluvium [13] and fractured metamorphic rocks with varying thickness ranging from <5 m to >50 m. The weight of 3 was assigned to the aquifer media, and the rating was assigned as per [6,22], as shown in Table 1.

### 2.2.4. Soil Media

Generally, the type of soil of a particular locality has an influence on groundwater contamination. Soil pollution potential largely depends on the grain size, the shrink and/or swell potential and the type and amount of clay present. These soil features have an influence on the purifying process of contaminants, the amount of water infiltrating into the ground and the amount of potential distribution. The surface and downward movements of contaminants are highly influenced by the soil cover characteristics. For example, the presence of fine-grained materials, such as silt, clay and organic matter within the soil, tends to lower the permeability, thus inhibiting contaminant migration through physico-chemical processes such as ionic exchange, biodegradation oxidation and absorption.

A soil map of the study area was prepared by downloading and extracting the digital soil map of the world in shapefile format from the Food and Agriculture Organization (FAO), where two groups of textural soil, namely loamy sand and silty clay, were found. Field observation also proved the texture of the soil in the study area. The weight of 2 was assigned to soil media, and the rating was assigned as per [6,22] as shown in Table 1.

### 2.2.5. Topography

This is an elevation of an area measured from the mean sea level. Areas with low elevation tend to retain water for a longer time and hence enable more percolation and infiltration of water, which may influence the movement of the contaminants into the aquifer. Steep slope areas are characterized by a large amount of runoff and small infiltration rates, hence less vulnerable to groundwater contamination, since a lot of contaminants may be washed away by running water.

In the present study, the topographic map was prepared from Digital Elevation Model (DEM) with 30 m by 30 m resolution, which was acquired from the United State Geological Survey (USGS). The DEM in meters was converted to slope using the slope function under the spatial analyst tool in Arc GIS. The weight of 1 was assigned for topography with rating values adopted from [6,22] as shown in Table 1.

### 2.2.6. Impact of the Vadose Zone

The vadose zone can be defined as an unsaturated zone that is found between the Earth's surface and the top of the aquifer zone. It is the position at which the groundwater is at atmospheric pressure. The lithological logging from Drilling and Dam Construction

Agency from 2017 to 2020 was studied to extract the thickness and the lithology of the vadose zone. Generally, the vadose zone of the study area was found to consist of sand with some patches of clay as well as gravel in some areas and weathered granitic gneiss materials. The impact of the vadose zone (I.V.Z.) was calculated as per [29] using Equation (2).

$$\text{I.V.Z} = (\,1/total\ depth) \times (\sum_{i=1}^{n} depth Li \times rating Li) \tag{2}$$

where *depth Li* is the depth of the particular lithological unit; *rating Li* is the rating of that particular unit, given by [6,26]; and total depth is the depth to the water table from the Earth's surface. The impact of the vadose zone is lower than the unit since it is a ratio. The greater the impact of the vadose zone, the higher the vulnerability to contamination. The impact of vadose was assigned the weight of 5, and the rating was modified from [6,22], as shown in Table 1.

### 2.2.7. Hydraulic Conductivity

This is the ability of the aquifer materials to convey water. The hydraulic conductivity of an aquifer depends considerably on the degree of saturation and intrinsic permeability. The higher the hydraulic conductivity of an aquifer, the higher the possibility of transmitting a large concentration of contaminants and vice versa. In the study area, the hydraulic conductivity was calculated from pumping test data (Cooper and Jacob solution) acquired from Drilling and Dam Construction Agency from 2017 to 2020. The hydraulic conductivity of the study area was in the range of 0.1 m/day–4.3 m/day. The hydraulic conductivity was assigned the weight of 3, and the rating was assigned as per [2]. as shown in Table 1.

### 2.2.8. Land Use/Land Cover

LU/LC can be described as the surface cover on the ground, such as vegetation and infrastructures as well as the human use of land in a particular place, which can represent cultural and economic activities such as agricultural, industrial, residential, recreational and mining uses. In the study area, the land use/land cover map was prepared by downloading a Landsat 8 image of Morogoro (path 167, row 65) from USGS website with 13.41% cloud cover that was taken on 26/8/2021. The image was classified in the ArcGIS environment using an interactive supervised classification function. The major LU/LC classes in the study area are settlement (build-up area), with minor areas falling under water bodies, bare land, agricultural land and vegetation cover, as shown in Figure 4. The weight of 5 was assigned to land use patterns with rating modified from [29], as shown in Table 1.

### 2.2.9. DRASTIC-LU/LC Index Map

The DRASTIC-LU/LC parameters (depth to water table, net recharge, aquifer media, soil media, topography, impact of vadose zone, hydraulic conductivity and land use/ land cover pattern) were reclassified under reclass function in the spatial analyst tool in ArcGIS. All parameters were reclassified into five classes, except for soil and aquifer media parameters, which have 2 and 3 classes, respectively. Each of the parameters was assigned a weight and rating based on the groundwater vulnerability potential. The decision on which parameter is more influential in groundwater contamination in relation to the other was aided by Analytical Hierarchy Process (AHP) as per [29,30]. The AHP tool was used to calculate the percentage of influence of each parameter, though field knowledge was key to final decision. Finally, the vulnerability index map was prepared under the weighted overlay function in ArcGIS based on an empirical formula, Equation (3), as applied by other researchers previously [2,4,5,29].

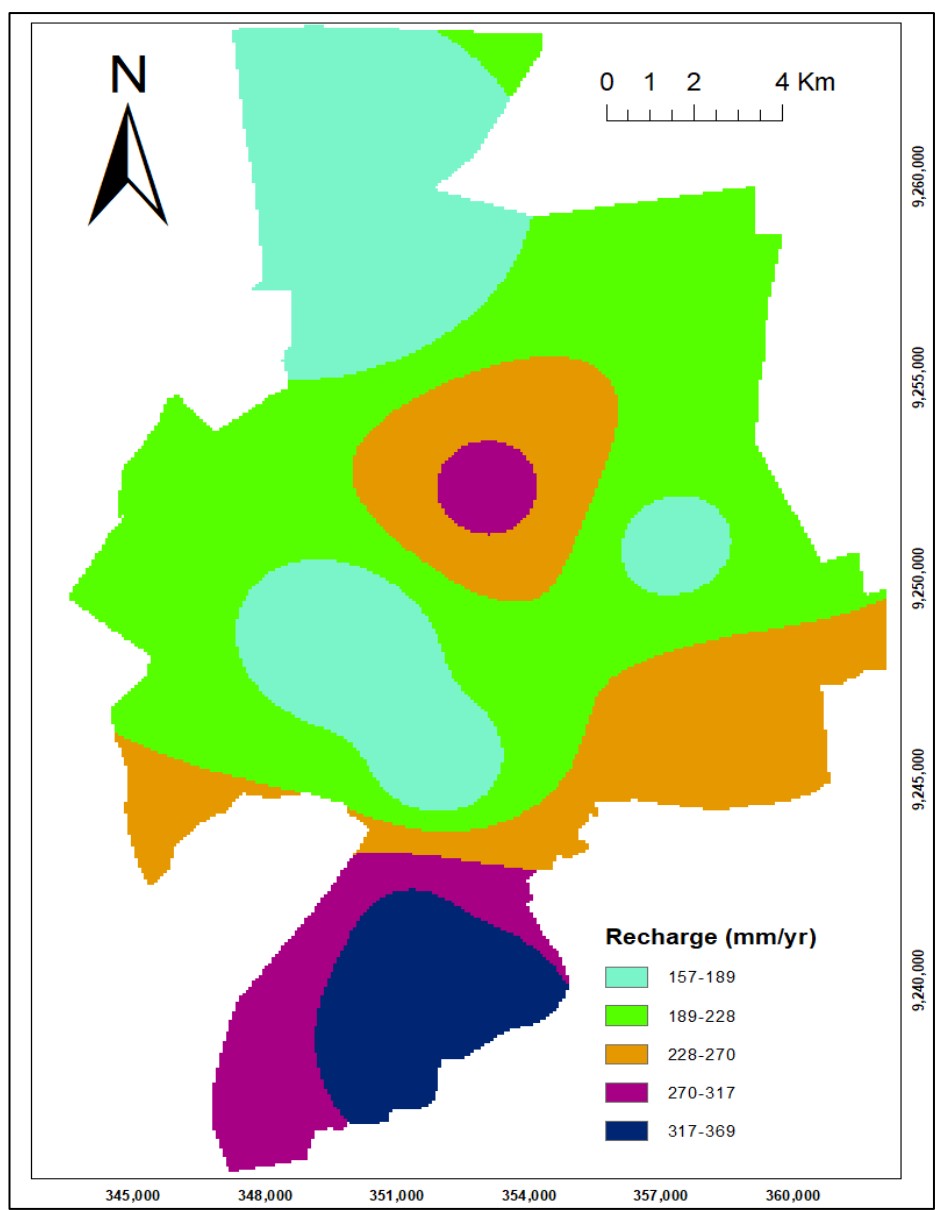

**Figure 4.** Net recharge map.

$$\text{DRASTIC} - \text{LU/LC index} = (Dr \times Dw) + (Rr \times Rw) + (Ar \times Aw) + (Sr \times Sw) + (Tr \times Tw) + (Ir \times Iw) + (Cr \times Cw) + (LU/LCr \times LU/LCw) \quad (3)$$

where $D$ = Depth to the water table, $R$ = Net recharge, $A$ = Aquifer media, $S$ = Soil media, $I$ = Impact of the vadose zone, $T$ = Topography, $C$ = Hydraulic Conductivity and *LU/LC* = Land use/Land cover. A relatively higher weight was assigned to land use/land cover, depth to water table and impact of vadose zone. This is because of the nature of the area under study, as the major land use type is urban settlement and hence the likelihood of releasing of contaminants to groundwater is high. Since the depth to water table is shallow in the large part of the study area (<10 m) and the vadose zone materials comprise of sand and gravel (high infiltration rate) (Table 2), the contaminants attenuation potential is low, and hence groundwater pollution potential is also high.

**Table 2.** Supervised classified land use confusion matrix.

| Class Value | Settlement | Agriculture | Vegetation | Water body | Bare land | Total | U_Accuracy | Kappa |
|---|---|---|---|---|---|---|---|---|
| Settlement | 18 | 0 | 0 | 0 | 2 | 20 | 0.9 | 0 |
| Agriculture | 0 | 14 | 6 | 0 | 0 | 20 | 0.7 | 0 |
| Vegetation | 0 | 0 | 20 | 0 | 0 | 20 | 1 | 0 |
| Water body | 0 | 0 | 0 | 20 | 0 | 20 | 1 | 0 |
| Bare land | 5 | 0 | 0 | 0 | 15 | 20 | 0.75 | 0 |
| Total | 23 | 14 | 26 | 20 | 17 | 100 | 0 | 0 |
| P-Accuracy | 0.78 | 1 | 0.77 | 1 | 0.88 | 0 | 0.87 | 0 |
| Kappa | 0 | 0 | 0 | 0 | 0 | 0 | 0 | 0.84 |

*2.3. Groundwater Sampling and Nitrate Analysis*

A total of forty (40) groundwater samples were collected for the analysis of nitrate in early March 2022 (a period of relatively low rainfall) using 500 mL prewashed, high-density polyethylene (HDPE) bottles. Water samples from boreholes and wells installed with water pumps were collected after pumping for a sufficient time to ensure that the stagnant water in the borehole was replaced by fresh water from the aquifer. On the other hand, the sampling of water in shallow wells with no motor pumps was carried out using a bailer. The samples were transported in ice-cold containers to the Department of Chemistry and Physics laboratory of the Sokoine University of Agriculture (SUA) for preparation and analysis of $NO_3^-$ concentration within 6 hours of collection. The onsite measurement of pH, temperature, electrical conductivity (EC) and total dissolved solids (TDS) was carried out during the sampling campaign using MM 156 Model pH/EC/T/TDS/DO Multi-parameter analyzer. The concentration of $NO_3^-$ was analyzed using the cadmium reduction method (DR 890) at the Department of Chemistry and Physics laboratory, SUA.

**3. Results and Discussion**

*3.1. DRASTIC-LU/LC Maps*

3.1.1. Depth to Water Table

From the water table depth map (Figure 5), shallow water table depth (0.6–4.9 m) was observed mostly in the southern part (mountain area). A large portion of the area, particularly the central area, falls into the medium water table depth (4.5–9.5 m). A relative high depth (9.5–15.2 m) was observed in the eastern and northern parts of the study area. The water table depth may influence groundwater contamination in two ways: First, if the aquifer is shallow, the contaminants reach the aquifer at a relative short time compared to the deep wells. On the other hand, in shallow wells, the oxic condition is common and thus nitrogenous contaminants can be oxidized to nitrate, while in deep wells, the anoxic condition influence reduction processes. With this in mind, the southern and the central part of the study area are more vulnerable to groundwater contamination than the other parts.

3.1.2. Net Aquifer Recharge

From the net recharge map (Figure 4), it can be observed that the southern area receives more rainfall and relatively high recharge as compared to the rest of the area. The relative high rainfall in the southern part of the study area is due to the influence of the Uluguru Mountains and the presence of thick vegetation. With respect to the recharge parameter, the southern area is therefore expected to be more vulnerable to groundwater contamination than the other parts of the study area.

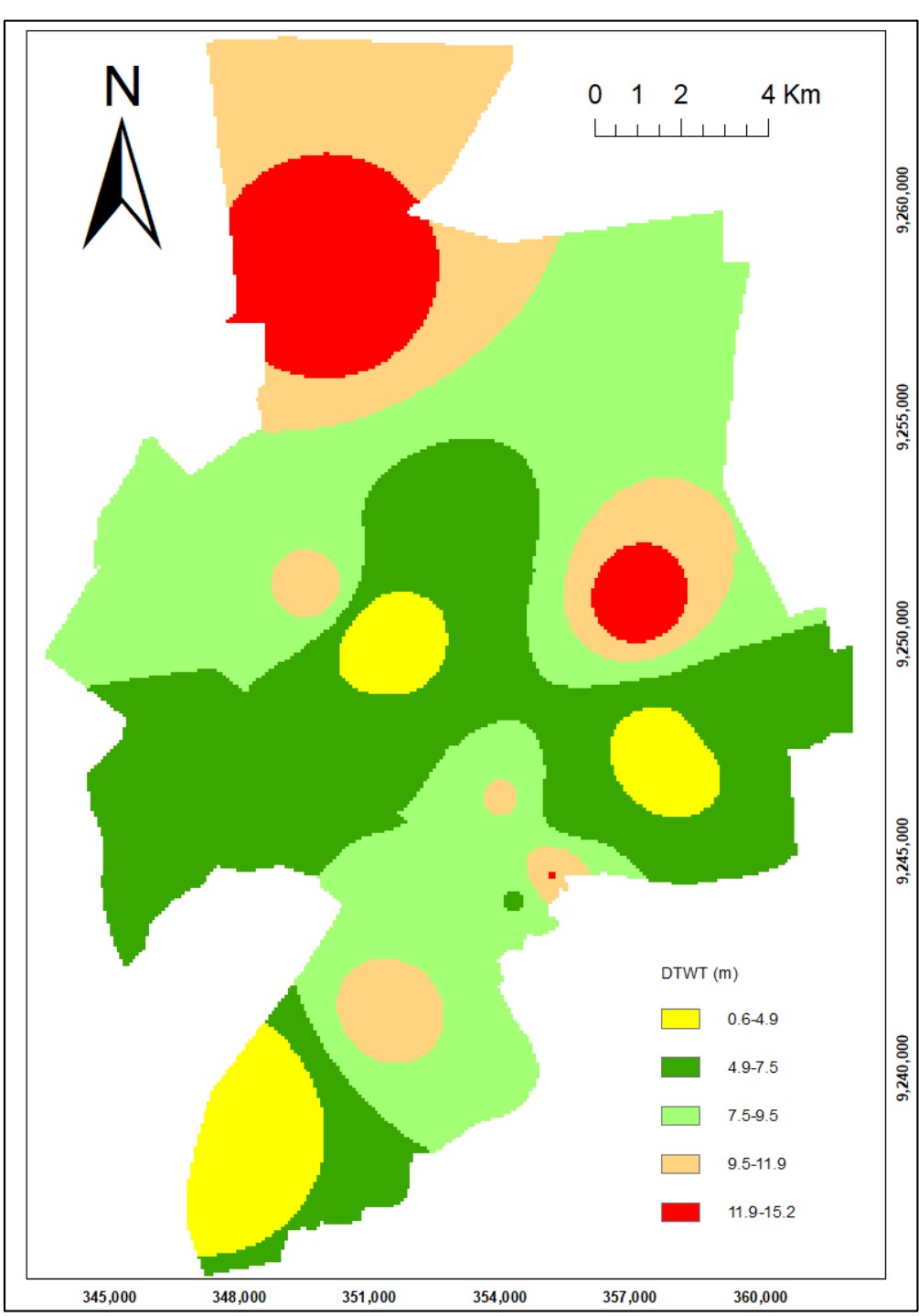

**Figure 5.** Depth to water table map.

### 3.1.3. Aquifer Media

The aquifer media (Figure 6) of the study area is made up of sand, alluvium and fractured metamorphic rocks (granulites and mica-migmatites). The central part of the study area comprises mainly of sediments (sandy and alluvium), while the southern part comprises of metamorphic rocks. Because the permeability and porosity are high in sedimentary rather than in metamorphic formations, the infiltration and percolation of contaminants are easier to occur into the aquifer through these formations. To that effect, the central part of the study area with sandy clay and alluvium is more vulnerable to groundwater contamination than the peripheral parts.

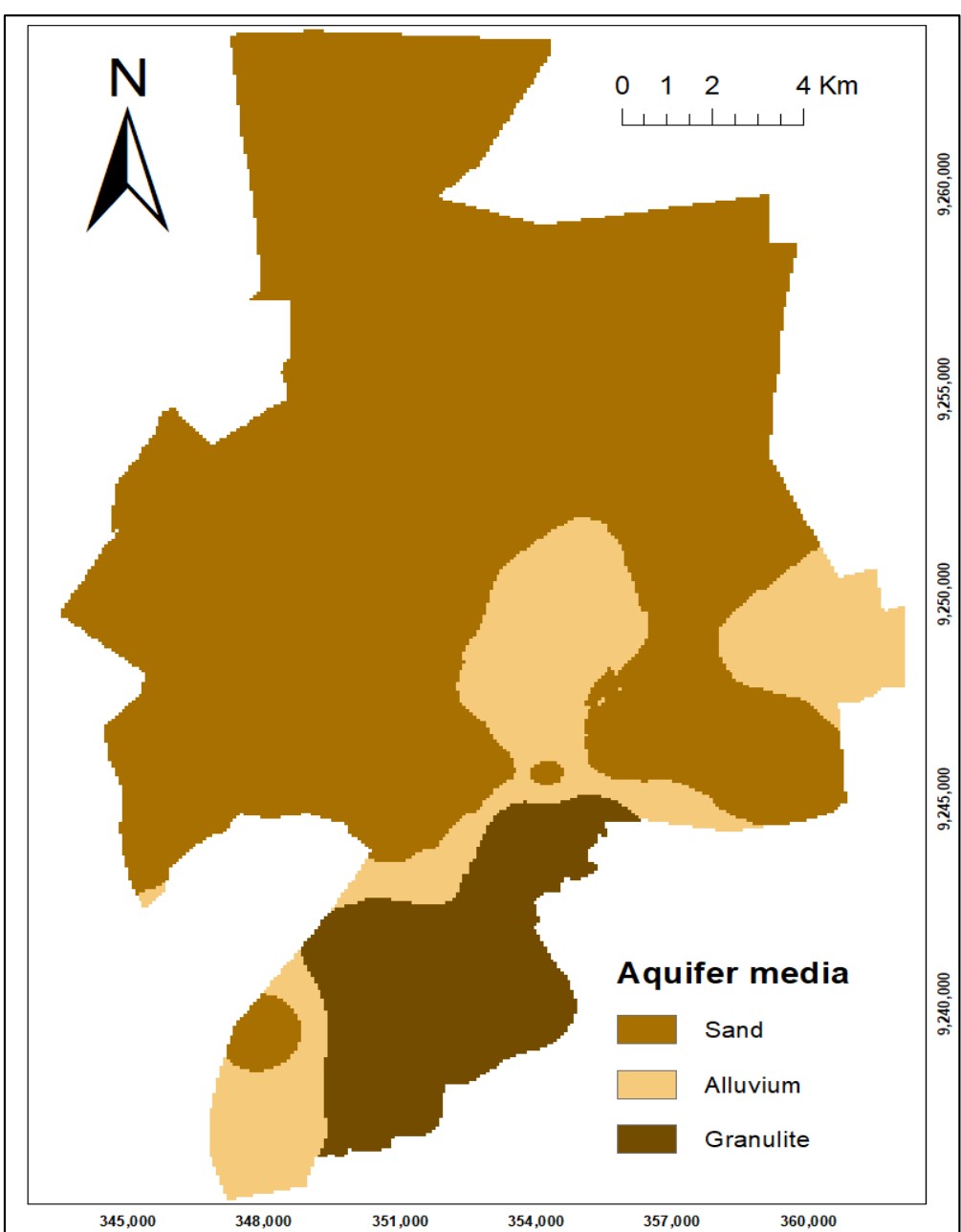

**Figure 6.** Aquifer media map.

### 3.1.4. Soil Media

From the soil media map (Figure 7), it can be observed that a large part of the study area, particularly the central part, comprises of silty clay soil, which has a low potential for groundwater contamination due to its low infiltration capacity and high-water retention [4]. The loamy sand soil (on the periphery of the study area) has a high infiltration rate and hence has a high groundwater vulnerability potential.

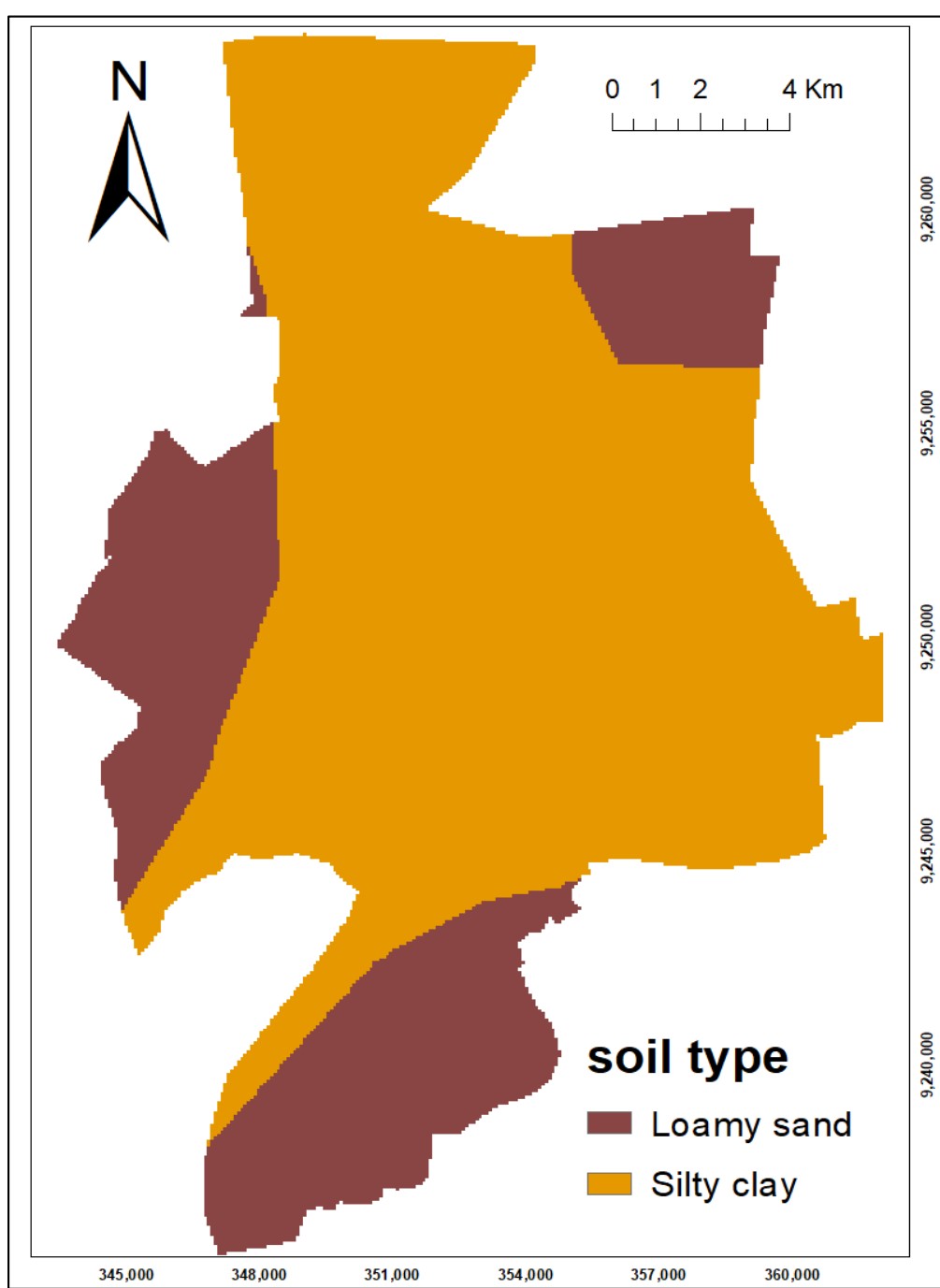

**Figure 7.** Soil media map.

### 3.1.5. Topography

The southern part in the topographical map (Figure 8) is found at a high elevation (slope >18%); thus, the runoff in this part is high relative to the flat areas in the central part. The central part of the study area allows water and contaminants accumulation and hence exposes the aquifer to groundwater contamination. Thus, based on the topography parameter, the southern part has a low vulnerability to groundwater contamination than the central area.

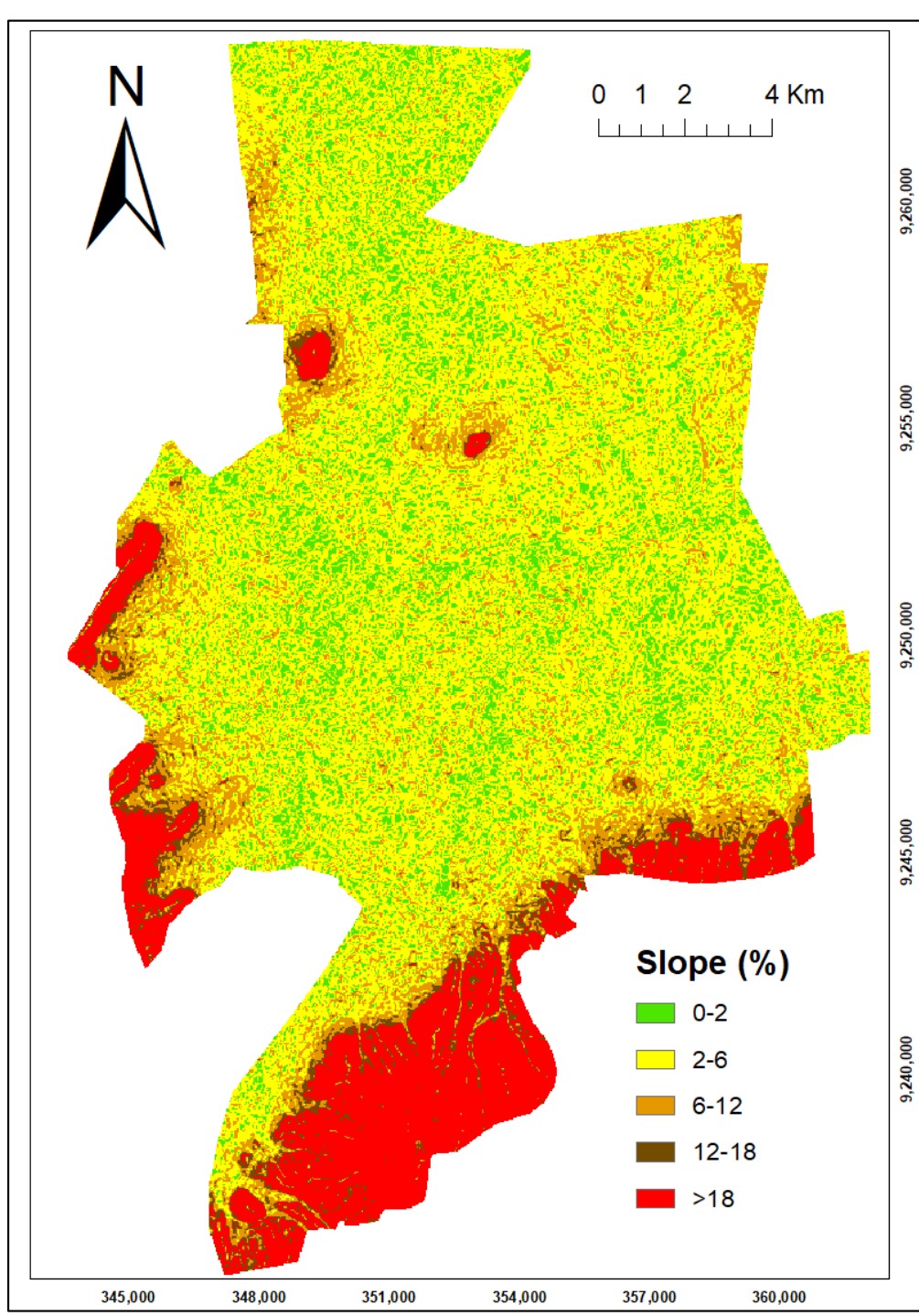

**Figure 8.** Topographical map.

### 3.1.6. The Impact of the Vadose Zone

The low impact of the vadose zone (Figure 9) was observed in the southern and northern parts of the study area, while a large portion, particularly the central part, was found in the moderate impact of the vadose zone. The lower the impact of the vadose zone, the higher the contamination risk, since the thickness of the vadose zone is low and thus the contaminants infiltrate into the aquifer easily and in a relatively short time.

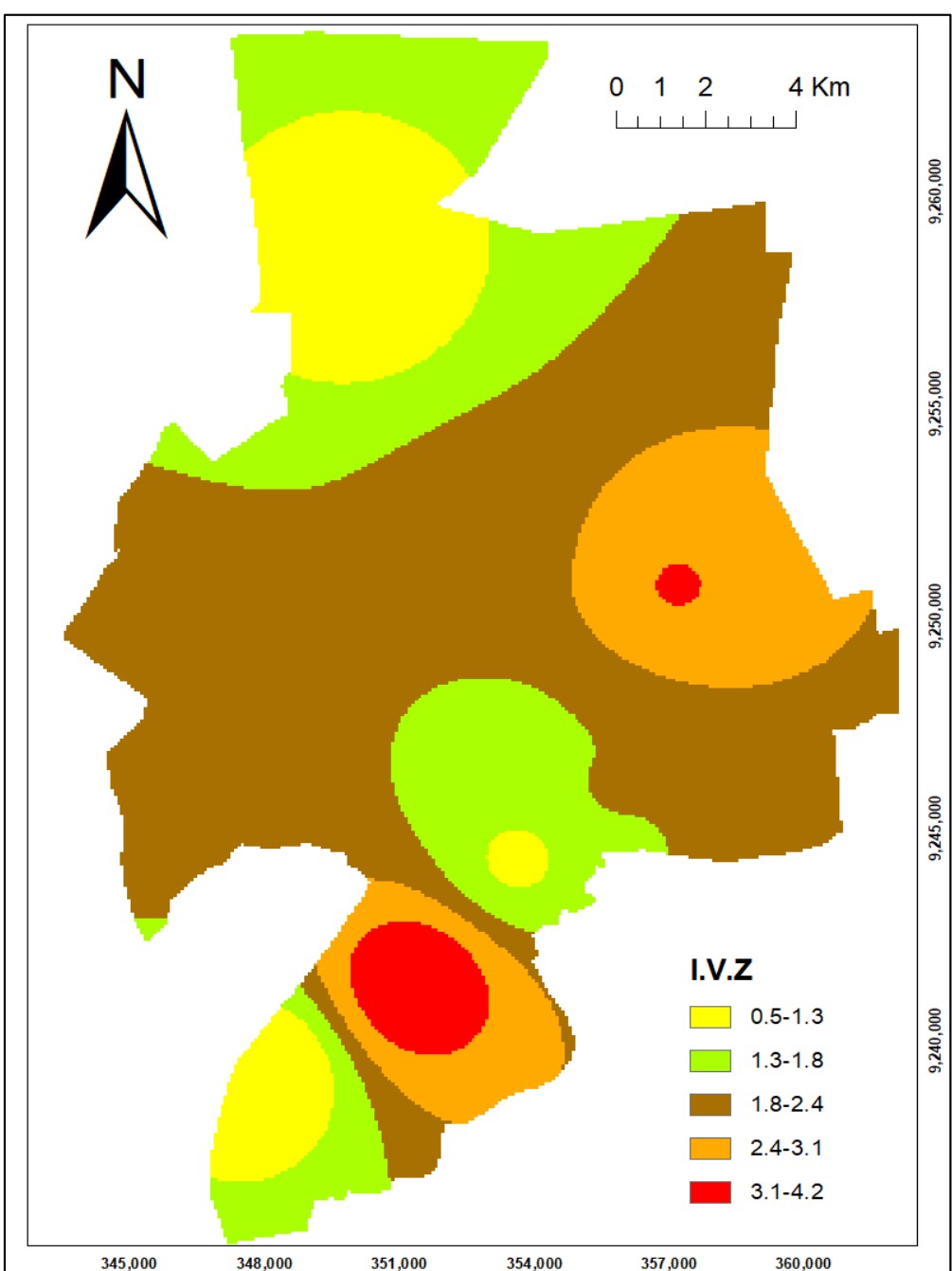

**Figure 9.** The impact of the vadose zone map.

### 3.1.7. Hydraulic Conductivity

A relatively high hydraulic conductivity (Figure 10) is found in the southern part of the study area and thus makes this part more vulnerable to groundwater contamination than the central part of the study area. Despite the southern part being dominated by metamorphic rocks, the conductivity was observed to be relatively high, possibly because of fractures that form the secondary porosity.

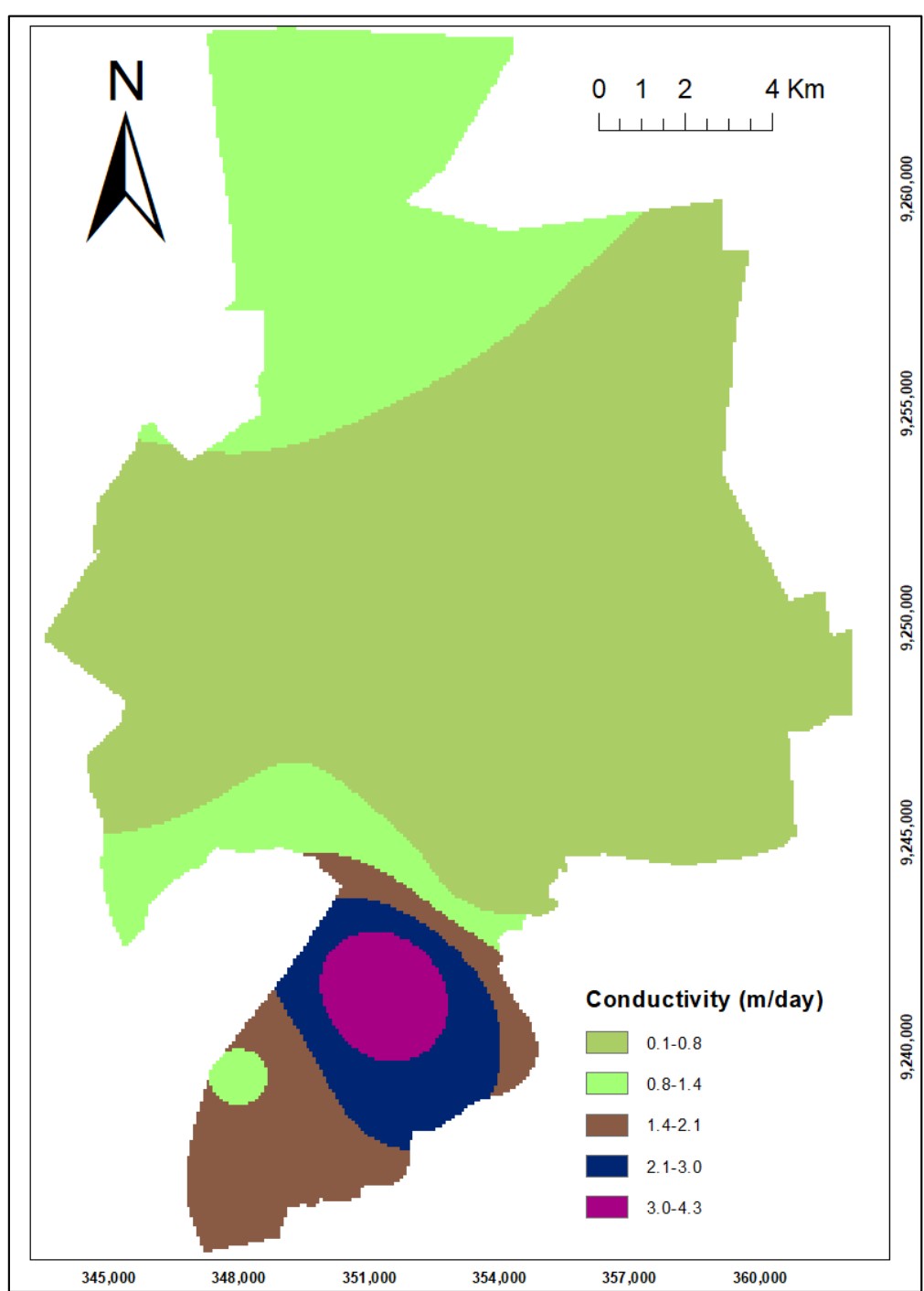

**Figure 10.** The hydraulic conductivity map.

### 3.1.8. Land Use/Land Cover

From the LU/LC map (Figure 11), about 70% of the area is classified as a settlement area. This implies that the area is likely to be contaminated because of the anthropogenic activities. According to [5,29–31], groundwater contamination in any place is highly triggered by anthropogenic activities, such as the use of organic and inorganic fertilizers and the poor design of latrine systems and septic tanks.

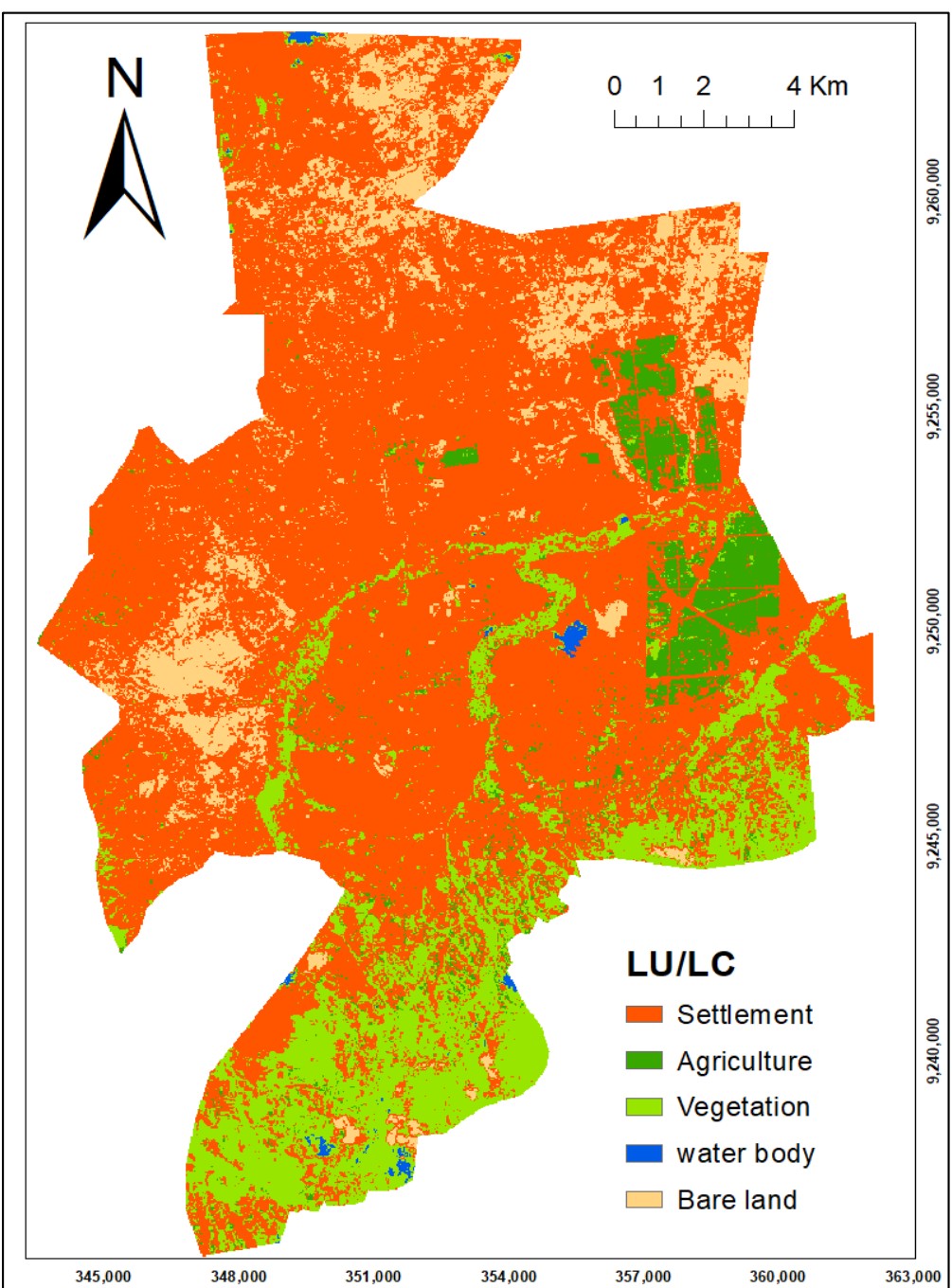

**Figure 11.** Land use/land cover map.

Accuracy Assessment

The LU/LC map (Figure 11) obtained from image classification may contain some errors. Thus, it is essential to assess the correctness of the results. The accuracy assessment of an image classification was performed by creating a classification confusion matrix, as shown in Table 2. Previous authors [32] proposed that the accepted accuracy assessment results should be at least 70%.

### 3.2. Vulnerability Index Map

The Analytical Hierarchy Process (AHP) with the aid of field observations was used in the multi-criteria decision to determine which parameter is more influential to groundwater contamination relative to the other. In this regard, the percentage influence of each

parameter to groundwater contamination was calculated (Table 3), and it was found that land use, depth to the water table, impact of the vadose zone and net recharge are the major parameters contributing to groundwater contamination within the study area more than aquifer media, soil media, topography and hydraulic conductivity. The same scenario was observed by previous researchers [5] in parts of the central Ganga Plain in India. From these percentages of influence, the DRASTIC-LU/LC model was computed in GIS and the classes were defined as shown in Figure 12.

**Table 3.** The percentage influence of the DRASTIC-LU/LC parameters.

| Parameter | % Influence (AHP) | Rating | Index |
|---|---|---|---|
| DTWT | 25 | 10<br>9<br>7<br>5 | 250<br>225<br>175<br>125 |
| Net recharge | 11 | 9<br>8<br>6 | 99<br>88<br>66 |
| Aquifer media | 5 | 8<br>6<br>3 | 15 |
| Soil media | 4 | 8<br>3 | 32<br>12 |
| Topography | 3 | 10<br>9<br>5<br>3<br>1 | 30<br>27<br>15<br>9<br>3 |
| I. Vadose zone | 21 | 8<br>6 | 168<br>126 |
| Conductivity | 5 | 10<br>8<br>6<br>4<br>1 | 50<br>40<br>30<br>20<br>5 |
| Land use | 26 | 10<br>8<br>5<br>3<br>1 | 260<br>208<br>130<br>78<br>26 |

The DRASTIC-LU/LC map was categorized into low, moderate and high-vulnerability zones with the respective areas of 29.2 km$^2$, 120.4 km$^2$ and 124.4 km$^2$. The central part of the study area falls into the high-vulnerability zone, being contributed to, mainly by the aquifer media (sandy and alluvium), the relatively shallow water table depth (<10 m), gently slope (<12%) and settlement land use pattern (dense habitation). The southern part of the study area falls into the moderate-vulnerability zone, despite having a relatively high net recharge and high vadose zone impact. This is because the area is found on a high elevation (slope > 18%) where there is less habitation and limited anthropogenic activities. The northern part of the study area is found in the low- to moderate-vulnerability zones due to high water table depth, low recharge, low vadose zone impact, clay soil type and relative low hydraulic conductivity as well as a large part of the area being bare land. The groundwater vulnerability to nitrate contamination is arguably associated with geochemical redox condition of the groundwater and depth of the aquifer [33]. Accordingly, in a high-vulnerability groundwater zone, the aquifer is relatively shallow and there is a possibility of a high concentration of nitrate ($NO_3^-$) and oxygen gas ($O_2$). In a moderate-vulnerability zone, the aquifer depth is moderate and the presence of oxidizing agents such as manganese ($Mn^{2+}$), Iron ($Fe^{3+}$) and sulphate ($SO_4^{2-}$) is common, while in the low-vulnerability groundwater zone, the aquifer depth is deep and the presence of the reducing agents such as hydrogen sulfide ($H_2S$), iron ($Fe^{2+}$) and methane ($CH_4$) gas is common.

The water temperature in the study area ranged between 25 °C and 33 °C, with a mean value of 29.33 °C and a standard deviation of 1.67 °C. The recommended temperature in water for various purposes should range from 20 °C to 35 °C (TBS, 2008). The groundwater temperature range in the study area were therefore in the permissible range. The pH of

groundwater in the study area ranged from 5.86 to 9.4, with a mean of 7.35 and a standard deviation of 0.64. Based on the standard from the World Health Organization (WHO, 2011) and Tanzania Bureau of Standards (TBS, 2008), the pH of water for drinking and other domestic uses should range from 6.5 to 8.5 and from 6.5 to 9.2, respectively. The findings show that pH values which deviate from the established standards were recorded at Msongeni (Bigwa, 5.86), slightly acidic, and riverside (Mwembesongo, 9.4), slightly alkaline. The decrease in pH favors the dissociation of natural rocks and induces chemical substance into groundwater, hence groundwater contamination [13].

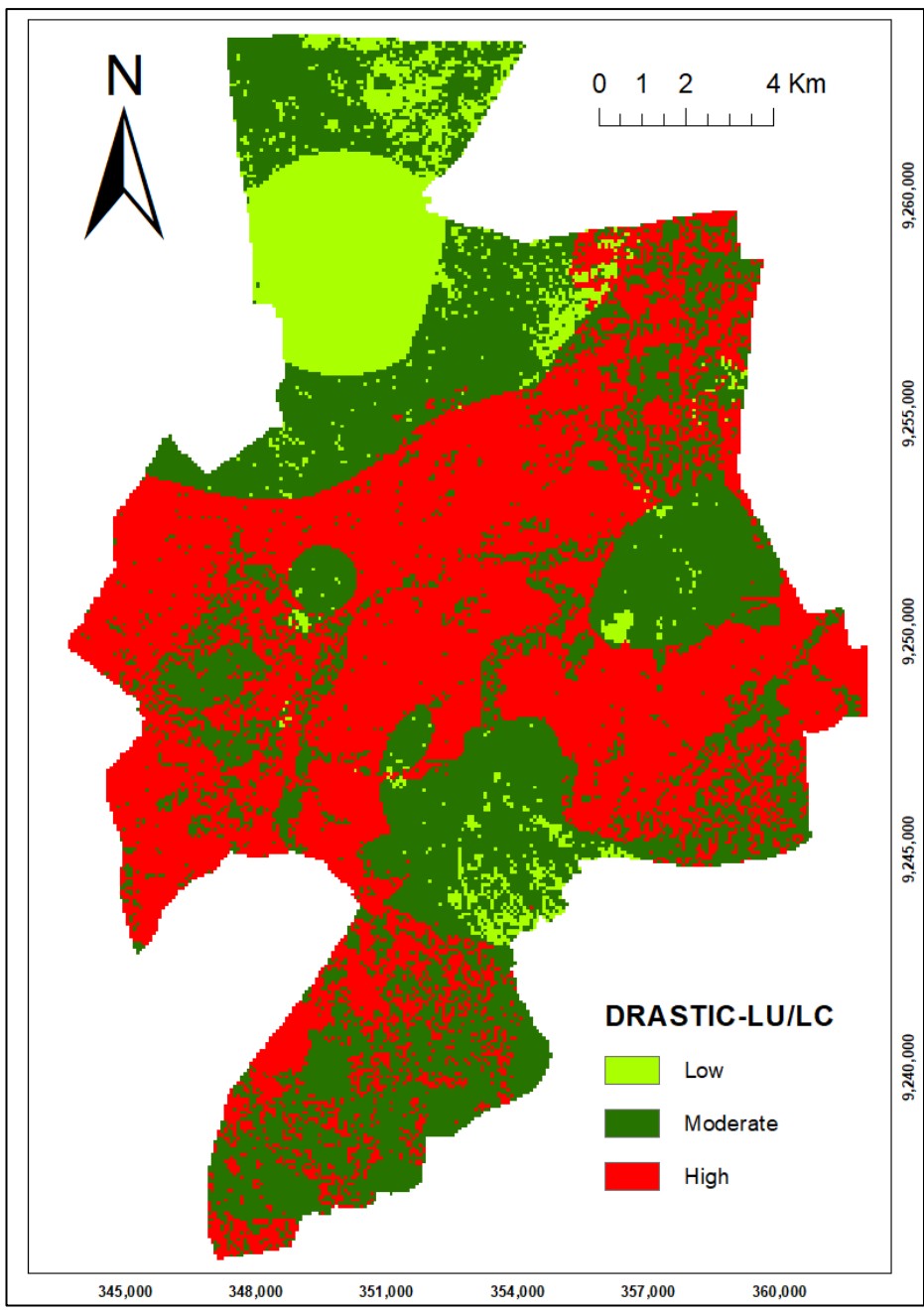

**Figure 12.** The DRASTIC-LU/LC model of the study area.

### 3.3. Nitrate Concentration in Groundwater

A total of forty groundwater samples were analyzed for nitrate and physico-chemical parameters, as presented in Table 4.

**Table 4.** Groundwater samples and their physico-chemical parameters and nitrate concentration.

| S/No | Location | Status | Temp °C | pH | EC (µs/cm) | TDS (mg/L) | Nitrate mg/L | Land Use |
|------|----------|--------|---------|------|------------|------------|--------------|----------|
| S1 | Kingolwira | BH | 27.2 | 6.67 | 1850 | 1017.5 | 35.7 | settlement |
| S2 | Kingolwira | SW | 28.2 | 7.26 | 1770 | 973.5 | 10 | agriculture |
| S3 | Kingolwira | BH | 29.1 | 6.5 | 4500 | 2475 | 222.7 | settlement |
| S4 | Bigwa | SW | 29.8 | 5.86 | 438 | 240.9 | 4.4 | agriculture |
| S5 | Bigwa | BH | 29.6 | 7.16 | 890 | 489.5 | 6 | settlement |
| S6 | Kilakala | BH | 32 | 6.91 | 1825 | 1003.75 | 12.7 | settlement |
| S7 | Kilakala | SW | 31.9 | 6.53 | 1657 | 911.35 | 68.6 | settlement |
| S8 | Kilakala | BH | 29.7 | 7.06 | 1440 | 792 | 20.9 | settlement |
| S9 | Mwembesongo | SW | 27.1 | 8.7 | 2100 | 1155 | 47 | settlement |
| S10 | Mwembesongo | SW | 29.9 | 9.4 | 1343 | 738.65 | 208.4 | settlement |
| S11 | Mwembesongo | SW | 29.7 | 7.85 | 2270 | 1248.5 | 48.1 | settlement |
| S12 | Kihonda | BH | 30.2 | 7.42 | 4250 | 2337.5 | 33.6 | settlement |
| S13 | Kihonda | DW | 31.4 | 8.35 | 2670 | 1468.5 | 64.1 | settlement |
| S14 | Kihonda | DW | 32.9 | 7.7 | 1082 | 595.1 | 89.4 | settlement |
| S15 | Mazimbu | DW | 30.4 | 7.8 | 3310 | 1820.5 | 208.4 | settlement |
| S16 | Mazimbu | SW | 28.7 | 7.71 | 1505 | 827.75 | 167.5 | settlement |
| S17 | Mazimbu | SW | 29.1 | 7.45 | 911 | 501.05 | 113.1 | agriculture |
| S18 | U/taifa | BH | 29.7 | 7 | 1435 | 789.25 | 284.1 | settlement |
| S19 | U/taifa | BH | 30.4 | 7.88 | 2170 | 1193.5 | 33.2 | settlement |
| S20 | U/taifa | SW | 28.1 | 8.33 | 1568 | 862.4 | 40.6 | settlement |
| S21 | Mafiga | BH | 28.8 | 6.71 | 1782 | 980.1 | 233.1 | settlement |
| S22 | Mafiga | BH | 29.2 | 7.58 | 985 | 541.75 | 98 | settlement |
| S23 | Mafiga | BH | 30.9 | 7.8 | 1881 | 1034.55 | 39.4 | settlement |
| S24 | Boma | BH | 31.6 | 7.64 | 616 | 338.8 | 40 | settlement |
| S25 | Boma | BH | 28.2 | 7.3 | 251 | 138.05 | 40 | settlement |
| S26 | Boma | BH | 30.2 | 7.5 | 611 | 336.05 | 9 | settlement |
| S27 | Mbuyuni | BH | 30.5 | 7.4 | 1642 | 903.1 | 109 | settlement |
| S28 | Mbuyuni | BH | 30.3 | 7.2 | 1978 | 1087.9 | 109 | settlement |
| S29 | Mbuyuni | DW | 28.7 | 7.2 | 868 | 477.4 | 49 | settlement |
| S30 | M/mpya | SW | 26.4 | 6.7 | 1043 | 573.65 | 98.2 | settlement |
| S31 | M/mpya | BH | 24.9 | 7 | 1985 | 1091.75 | 43.4 | settlement |
| S32 | M/mpya | SW | 26.4 | 6.5 | 862 | 474.1 | 38.3 | settlement |
| S33 | Kichangani | BH | 27 | 7.5 | 4190 | 2304.5 | 248.8 | settlement |
| S34 | Kichangani | BH | 27.9 | 7.1 | 654 | 359.7 | 76 | settlement |
| S35 | Kichangani | DW | 28.8 | 7.2 | 200 | 110 | 24 | agriculture |
| S36 | Bigwa | BH | 29.6 | 7.3 | 927 | 509.85 | 222.8 | settlement |
| S37 | Bigwa | SW | 28.2 | 6.8 | 1207 | 663.85 | 142 | settlement |
| S38 | Kingo | BH | 30.2 | 7.5 | 964 | 530.2 | 5.4 | settlement |
| S39 | Kihonda | BH | 30.4 | 7.2 | 1786 | 982.3 | 14.8 | settlement |
| S40 | Kihonda | BH | 29.8 | 7.4 | 2160 | 1188 | 30 | settlement |

BH: Bore hole (>30 m), SW: Shallow well (15–30 m) and DW: Dug well (<15 m).

The electrical conductivity of water is directly related to the concentration of dissolved ions in water; thus, a high value of EC implies that more ions are dissolved in water. The EC was measured in the field immediately after water sampling because conductivity changes with time and depends on temperature. The electrical conductivity of the groundwater ranged from 200 to 4500 (µs/cm), with a mean value of 1639.4 µs/cm and a standard deviation of 1015.24 µs/cm. The recommended electrical conductivity for domestic water uses is 1400 µs/cm and 2000 µs/cm, based on WHO, (2011) and TBS (2008) standards, respectively. From these standards, it is observed that 22.5% and 57.5% of groundwater samples in the study area deviate from TBS and WHO standards, respectively, an indication of the presence of more dissolved solids in groundwater.

Similarly, the total dissolved solids (TDS) ranged from 110 mg/L to 2475 mg/L with a mean value of 901.67 mg/L and a standard deviation of 558.38 mg/L. TDS in water indicates the presence of salts and inorganic matter in water. The chief elements are potassium, chloride, magnesium, nitrate, sodium, calcium, carbonate, hydrogen carbonate and sulphate, though these elements were not analyzed in this study. The recommended amount of TDS by TBS (2008) and WHO (2011) should be in the range of 500–600 mg/L and 500 mg/L, respectively. It was observed that 57.5% of groundwater samples have TDS above the permissible limit (Table 4).

Nitrate concentration in groundwater ranged from 4.4 to 284.1 mg/L with a median value of 47.5 mg/L and a standard deviation of 79.3 mg/L. Based on the WHO (2011) and TBS (2008) standards, the concentration of nitrate in groundwater for drinking should not exceed 50mg/L. Thus, 45% of the groundwater samples had a nitrate concentration beyond the permissible level, and an extremely nitrate concentrations (>200 mg/L) were

recorded at Kingolwira, Mwembesongo, Mazimbu, Uwanja wa Taifa, Mafiga and Bigwa wards. According to previous studies [12], the background concentration of nitrate (from rock source) in Tanzania has been established at 2.5 mg/L. Thus, the elevated concentration of nitrate in groundwater above the background concentration is highly influenced by anthropogenic activities taking place in the study area. This was proved during fieldwork, where most of the septic tanks in different places were observed to be installed nearby wells with less consideration on the effect of elevation. Furthermore, the ongoing urban farming activities (e.g. home gardens) associated with the use of fertilizers may be another factor for the observed high nitrate concentration in groundwater. The spatial distribution of nitrate concentration was provided with the risk classification (Figure 13) modified from [2].

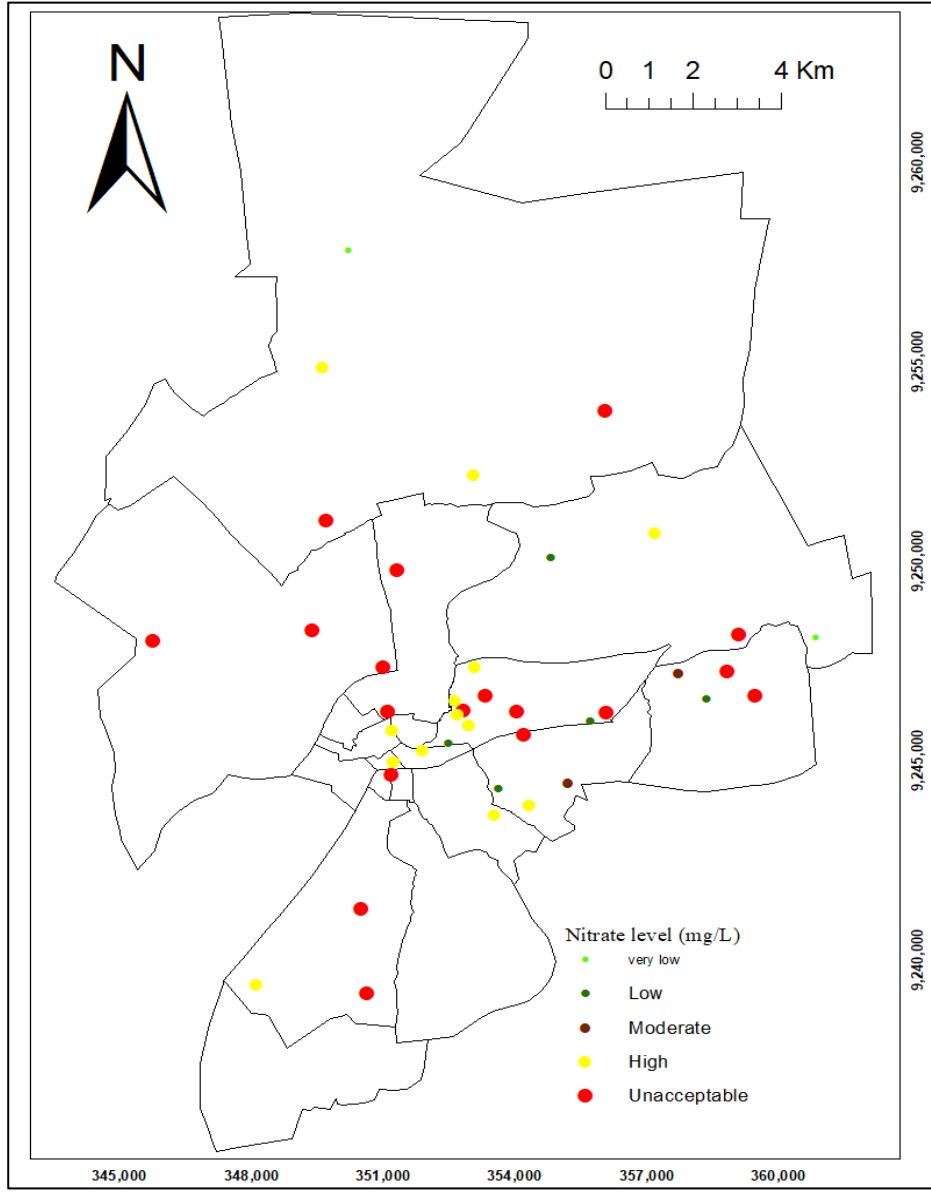

**Figure 13.** Spatial distribution of nitrate and classification map: very low (<5 mg/L), low (5–13 mg/L), moderate (14–28 mg/L), high (29–50 mg/L) and unacceptable (>50 mg/L).

## 4. Model Validation Using Experimental Results

The DRASTIC-LU model was validated by using the concentration of nitrate from thirty-three groundwater samples that were spatially distributed within the municipality. It was found that 55% (10 samples out of 18), 15% (2 samples out of 13) and 50% (1 sample out of 2) with unacceptable, high and moderate nitrate concentrations, respectively, fall into the

high-vulnerability zone. Furthermore, 45% (8 samples out of 18), 70% (10 samples out of 13) and 50% (1 sample out 2) with unacceptable, high and moderate nitrate concentrations, respectively, fall into the moderate-vulnerability zone. In the low-vulnerability zone, only 15% (1 out of 13) of the samples had a high nitrate concentration, as shown in Figure 14.

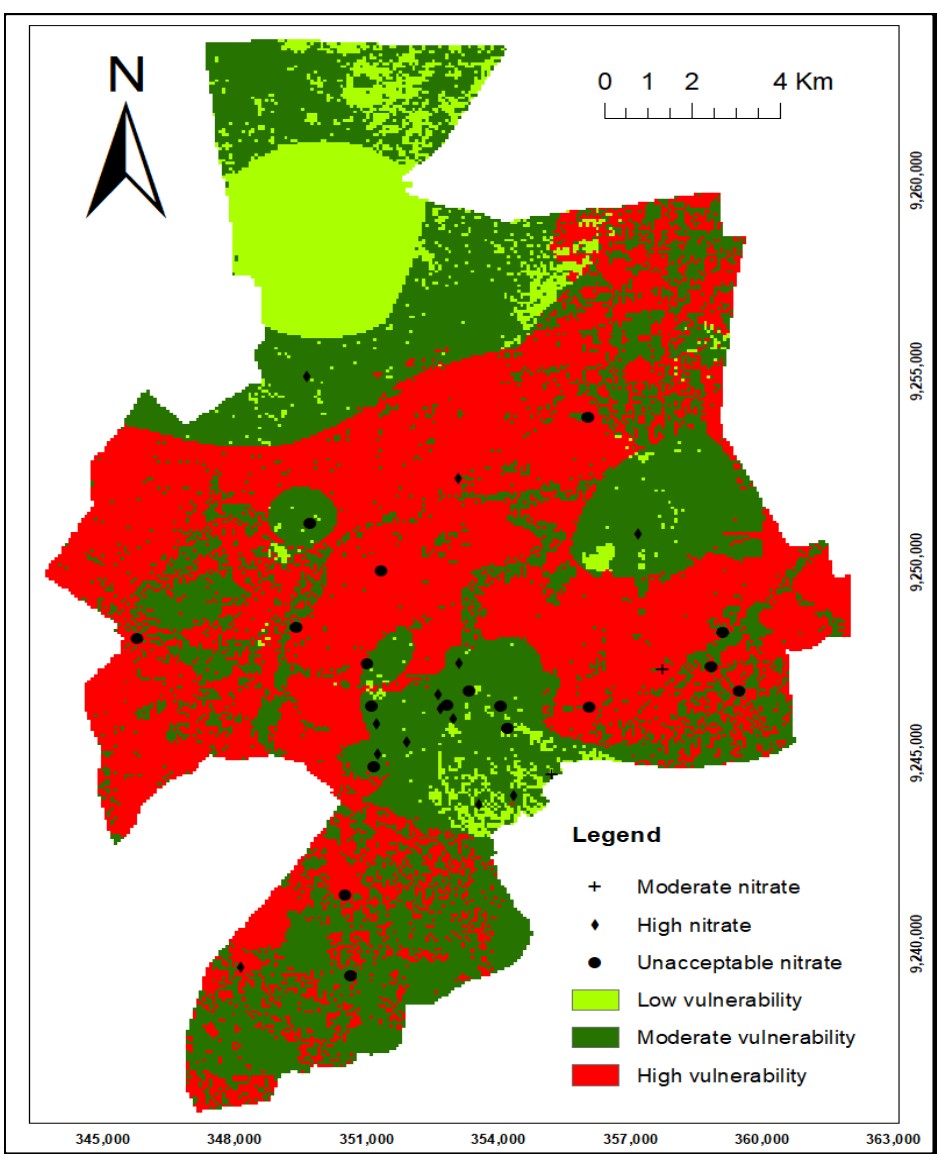

**Figure 14.** DRASTIC-LU/LC model and nitrate concentration.

## 5. Conclusions

Groundwater vulnerability mapping is very important within the study area and the country in general. This is because groundwater resource continues to be the trusted alternative water source in addition to surface water sources that are highly exposed to both natural (climate change) and anthropogenic (land use) forces. With the use of DRASTIC-LU/LC model in a GIS environment, the groundwater vulnerable areas in Morogoro municipality were delineated. This was achieved by preparing eight parametric maps and overlaying them to generate the vulnerability index map with the aid of various GIS functions. The DRASTIC-LU/LC Vulnerability Index map was classified into three zones, namely, low-vulnerability zone (area = 29.2 km$^2$), moderate-vulnerability zone (area = 120.4 km$^2$) and high-vulnerability zone (area = 124.4 km$^2$), which mark the percentage values of 10.6%, 44% and 45.4%, respectively. The major contributing factors for groundwater contamination in the study area are land use pattern, depth to water level

and the impact of the vadose zone, with land use patterns being the leading contributing factor since it acts as the source of contaminants during the use of manure and fertilizers in farming activities as well as the use of latrine and sewage systems. However, this scientific predicament can be better proved by the use of stable isotopes of nitrate.

**Author Contributions:** Conceptualization, N.J.M.; methodology, N.J.M., I.C.M., K.R.M. and E.E.M.; software, N.J.M.; validation, N.J.M., I.C.M., K.R.M. and E.E.M.; formal analysis, N.J.M., K.R.M., I.C.M. and E.E.M.; investigation, N.J.M.; resources, N.J.M., K.R.M., I.C.M. and E.E.M.; data curation, N.J.M., K.R.M., E.E.M. and I.C.M.; writing—original draft preparation, N.J.M.; writing—review and editing, I.C.M., K.R.M., E.E.M. and N.J.M.; visualization, N.J.M., K.R.M., I.C.M. and E.E.M.; supervision, I.C.M., E.E.M. and K.R.M. All authors have read and agreed to the published version of the manuscript.

**Funding:** This research was funded by The University of Dodoma and the APC was funded by the authors.

**Institutional Review Board Statement:** Not applicable.

**Data Availability Statement:** Not applicable.

**Acknowledgments:** The author wishes to extend her sincere thanks to the University of Dodoma for the sponsorship, the supervisors for technical assistance, Wami–Ruvu Water Basin Management and Tanzania Meteorological Agency (TMA) for data assistance, and the family for the courage and love.

**Conflicts of Interest:** The authors declare no conflict of interest. The funders had no role in the design of the study; in the collection, analyses, or interpretation of data; in the writing of the manuscript; or in the decision to publish the results.

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
