# Peer review of "The Use of the DRASTIC-LU/LC Model for Assessing Groundwater Vulnerability to Nitrate Contamination in Morogoro Municipality, Tanzania"

_2673-4834, doi:10.3390/earth3040067_

Round 1

Reviewer 1 Report

Dear authors

To improve the present manuscript, consider the following comments carefully.

1- Materials and methods are not clearly stated in the abstract section.

2- Some parts of introducing sentences are stated without reference (lines 48-45). Use the following references to improve the introduction and other sections.

Optimization of statistical and machine learning hybrid models for groundwater potential mapping

Anthropogenic nitrate in groundwater and its health risks in the view of background concentration in a semi arid area of Rajasthan, India

Machine learning predictions of nitrate in groundwater used for drinking supply in the conterminous United States

A high-resolution nitrate vulnerability assessment of sandy aquifers (DRASTIC-N)

3- Add more information about the study area—type of land use, type of geological formations, the climate of the studied area, etc.

4- Make the text font smaller in all figures. The text should not be placed on the maps.

5- Why do settlement areas have the most weight in Table 2? What do these weights represent?

6- Are tables 2 and 3 the results of your work or part of the materials and methods?

7- The quality of the figures is low. Make the figure legend font smaller. In the figures, write the title of each figure instead of the LEGEND word.

8- Only a few figures are given in the results section. This section should be more detailed.

9- Figures and tables are usually not given in the discussion section. I think you should merge the results and discussion sections together.

Author Response

Hi! reviewer 1

Please find the attached document comprises responses of your comments and suggestions

Reviewer 2 Report

 The research is interesting and here are the comments for the Text:

The abstract contains % in two places not lead to 100% (check)

In the Introduction:

-        In the Introduction (the first three lines) The first reference of 99% about groundwater fresh water is not correct. The same for the first paragraph of section 4.4 (The percent is unclear)

-        For the second reference about India (80% and 50 %) is not completely mentioned (rewrite the sentence).

-        In the location area there is mixing between Dar Es lasm city and Dodoma

-        In climate: the average rainfall is one number and varying from ---to ---(P4 Line 103)

-        In Net recharge: Why the authors use the Thornthwait Equations and not Pennman (Any differences) (P6 ;line 150)

-        The aquifer media calculation is not clear (P6 line 167)

-        Line 231 Study instead of stud

-        New section should be added 2.2.10  Nitrate sampling and contains the first paragraph of P10

-        The section Results has no data (Add data and text)

-        Section 4.1 (rewrite) in order to have a complete picture of the research

-        Rewrite the conclusion

Author Response

Hi! reviewer 2

Please find the attached document comprises responses of your comments and suggestions. The responses were given point by point and the track changes mode was activated in the manuscript.

Reviewer 3 Report

The manuscript is very interesting, the authors have described the usage of the DRASTIC-LU/LC model of groundwater vulnerability.  

Reading the manuscript, the question arises whether it is an original modification of the DRASTIC method by the authors or it has already been used elsewhere. If it is an original method, it is necessary to change the title of the article.

 To achieve better quality of the manuscript, I suggest the following:

Chapter 1.

Line 54-56 – lots of data about the nitrate concentrations in several cities, but with no reference. Please add the reference for that data

Chapter 2.

All localities mentioned in the manuscript must be shown on the location sketch (line 93; 112-114,…)

Figure 1 is not legible, especially the municipalities in the upper left picture. Distribution of sample points is OK, but roads and rivers without toponyms do not contribute to the readability of the image

Figure 2 – change the “period of 15 years” with real period (2007-2021); reduce the size of the figure, such a large figure is not needed in the manuscript

Line 111-112 – latitude and longitude is not necessary in the manuscript

Considering that DRASTIC requires a large number of geological and hydrogeological parameters, a better hydrogeological description of the research area is needed, possibly a hydrogeological profile if applicable

DDCA, 2017-2020 – written as reference but it is not. If it is published data, treat it as a reference and put it in the reference list, but if it is raw data, then without abbreviations put from whom the data were taken and for what period

Figure 3 - top left image blurry and unnecessary; the main image looks more like a geological map. On the hydrogeological map, it is necessary to highlight the permeability of rocks and deposits, catchment areeas, water divide, springs, water supply objects,...

Line 138 - it is unnecessary to mention the software and the manufacturer of the software in the scientific article

Table 2 - the range of the conductivity parameter indicates that the method was developed only for this area, as the text states that the conductivity is in the range 0.1-4.3. In that case, this method is not applicable in any other area because it is unlikely that the bandwidth range will be exactly in that range.

Chapter 4.

Chapter 4.3. move from the chapter 4. Discussion to the chapter 3. Results

Figure 6 & 7– reduce the sizes of the figures

Line 380-381 - remove the toponyms from the manuscript or show them on one of the figures.

Figure 8 - almost all samples with unacceptable nitrate concentrations are located in areas of low or moderate vulnerability. Explain in the discussion.

Author Response

Hi! reviewer 3

Please find the attached document comprises responses of your comments and suggestions. The responses were given point by point and the track changes mode was activated in the manuscript.

Round 2

Reviewer 1 Report

This article can be published in this format.

Author Response

Hello reviewer 1

Thank you for your approval of this manuscript

Reviewer 2 Report

The Text is improved and the authors gave explanation of all raised points during the first review, some comments are:

Page 2 line 56 previous studies is not popular, add exact reference

Organize the paragraphs in P 25

Author Response

Hello reviewer 2,

Please find the responses of your comments

Reviewer 3 Report

Point 1: Figure 2 –reduce the size of the figure, such a large figure is not needed in the manuscript

Author Response

Hello reviewer 3,

Please find the response of your comment
